# Indicator of Inflammation and NETosis—Low-Density Granulocytes as a Biomarker of Autoimmune Hepatitis

**DOI:** 10.3390/jcm11082174

**Published:** 2022-04-13

**Authors:** Weronika Domerecka, Iwona Homa-Mlak, Radosław Mlak, Agata Michalak, Agnieszka Wilińska, Anna Kowalska-Kępczyńska, Piotr Dreher, Halina Cichoż-Lach, Teresa Małecka-Massalska

**Affiliations:** 1Department of Human Physiology, Medical University of Lublin, 11 Radziwiłłowska Str., 20-080 Lublin, Poland; iwona.homa.mlak@gmail.com (I.H.-M.); radoslaw.mlak@gmail.com (R.M.); teresamaleckamassalska@umlub.pl (T.M.-M.); 2Department of Gastroenterology with Endoscopy Unit, 8 Jaczewskiego Str., 20-090 Lublin, Poland; lady.agatamichalak@gmail.com (A.M.); halina.lach@umlub.pl (H.C.-L.); 3Department of Clinical Genetics, Medical University of Lublin, 11 Radziwiłłowska Str., 20-080 Lublin, Poland; agnieszka.wilinska@umlub.pl; 4Department of Biochemical Diagnostics, Laboratory Diagnostics, Medical University of Lublin, 16 Staszica Str., 20-081 Lublin, Poland; annakowalskakepczynska@umlub.pl; 5Public Health, Medical University of Lublin, 1 Chodźki Str., 20-093 Lublin, Poland; piotr.dreher@umlub.pl

**Keywords:** autoimmune hepatitis, LDG, MPO, NET, NETosis, inflammation

## Abstract

Introduction. Interest in the potential role of low-density granulocytes (LDGs) in the development of autoimmune diseases has been renewed recently. Due to their pro-inflammatory action, more and more attention is paid to the role of LDGs, including those expressing the enzyme myeloperoxidase (MPO), in the development of autoimmune hepatitis (AIH). LDGs are actively involved in the formation of neutrophil extracellular traps (NETs). This phenomenon may favour the externalization of the autoantigen and lead to damage to internal organs, including the liver. Aim. The main aim of the study was to assess the diagnostic usefulness of the LDG percentage, including the fraction showing MPO expression as markers of systemic inflammation in AIH. Materials and methods. The study included a group of 25 patients with AIH and 20 healthy volunteers. Mononuclear cells, isolated from peripheral blood, were labelled with monoclonal antibodies conjugated to the appropriate fluorochromes (CD15-FITC, CD14-PE, CD10-PE-Cy5, MPO+) and then analyzed on a Navios Flow Cytometer (Beckman Coulter). Results. Patients with AIH had a higher median percentage of LDG (1.2 vs. 0.1; *p* = 0.0001) and LDG expressing MPO (0.8 vs. 0.3; *p* = 0.0017) when compared to healthy volunteers. Moreover, the percentage of LDG was characterised by 100% of sensitivity and 55% of specificity (AUC = 0.84; *p* < 0.0001), while the percentage of LDG expressing MPO was 92% of sensitivity and 55% of specificity (AUC = 0.78; *p* = 0.0001) in the detection of AIH. Conclusions. Assessment of inflammatory markers, such as the percentage of LDG and the percentage of LDG expressing MPO, may be helpful in assessing the phenomenon of an increased systemic inflammatory response and in assessing liver fibrosis (LC, Liver cirrhosis), which is inherent in liver decompensation. Taking into account the above arguments, the assessment of the percentage of LDG, including LDG expressing MPO, may turn out to be a useful marker in the diagnosis of AIH.

## 1. Introduction

Autoimmune hepatitis (AIH) is a chronic, immune-mediated disease with a prevalence of 10 to 17 per 100,000 people in Europe [1]. AIH may be asymptomatic or present in various forms, ranging from subclinical disease to acute and end-stage liver failure [2]. The diagnosis of AIH is based on the presence of specific antibodies, including antinuclear antibodies (ANA), anti-smooth muscle antibodies (SMA), liver/kidney microsome type 1 antibodies (anti-LKM1), and anti-soluble liver antigen/liver pancreas (anti-SLA) with increased concentration of immunoglobulin G (IgG) in the serum. The treatment is based on corticosteroids and immunosuppression. Hepatic transplantation is performed in patients with severe liver failure in the course of acute AIH and in patients with already developed liver cirrhosis (LC), followed by hepatocellular carcinoma [3]. The inflammatory process in the course of AIH is systemic in nature. The onset of the disease is probably associated with impaired T lymphocyte function [4], the development of molecular mimicry [5], intestinal dysbiosis [6] and infiltration of low-density neutrophils (LDG) [7], which show the features of reactive cells and, together with autoantibodies (i.e., ANA, ASMA), imply the formation of Neutrophil Extracellular Traps (NETs). NETs are produced by a process of active cell death called NETosis. The development of NETs, which determines an abnormal immune response and thus the intensification of inflammation and tissue damage, has been observed, among others, in atherosclerosis, systemic lupus erythematosus (SLE) and rheumatoid arthritis (RA) [8]. The absolute number of chronic liver disease (CLD) patients (regardless of their severity) accounts for 1.5 billion cases worldwide. LC constitutes a final stage of CLD due to multiple factors (i.e., non-alcoholic fatty liver disease (NAFLD), alcohol-related disease, hepatitis B or C infection, autoimmune diseases, cholestatic disorders, and iron or copper overload). LC develops after a long-lasting period of inflammation, resulting finally in a reversible replacement of the healthy hepatocytes with fibrotic tissue and regenerative nodules, leading to the development of portal hypertension. The management of LC aims at struggling with the causes and complications, while in some cases liver transplantation is needed [9].

Two populations of cells are observed in the peripheral blood, i.e., polymorphonuclear leukocytes (PMN) and peripheral blood mononuclear cells (PBMC). LDG are neutrophilic granulocytes which, after separation in the density gradient, remain in the PBMC fraction [10]. When first described, LDG was found characteristic of rheumatic diseases, such as SLE [11]. More recent studies describe the presence of LDG in various diseases, such as asthma [12], tuberculosis [13], arthritis [14], cancer [15], sepsis [16], and HIV [17]. They have also been observed in animal models of viral infection, such as classical swine fever (CSF) virus [18]. In addition, despite the above-mentioned SLE, LDG has been observed in various autoimmune diseases, including psoriasis [19] and RA [20]. Nevertheless, according to the available literature, the above-mentioned cell subsets have never been investigated among AIH patients before. Therefore, due to the inflammatory background of AIH, this disease may be perceived as a good entity to explore any potential role of selected immune cells in its natural history.

Similar to mature normal-density granulocytes (NDGs), LDGs strongly express the characteristic surface CD15, CD10, CD11b, CD11c, and CD16 markers, as detected by flow cytometry. However, unlike NDGs, the shape of the nuclei in LDGs indicate younger forms of developmental series of granulocytes [21,22]. In practice, an appropriate density gradient cell separation medium is additionally used to distinguish LDG from NDG. LDGs demonstrate the features of pro-inflammatory cells. When activated, they can damage endothelial cells and release a large amount of tumour necrosis factors (TNF) and type I and II interferons (IFN) [8,23,24]. Activated by pathogenic microorganisms or pro-inflammatory cytokines, LDGs can undergo spontaneous NETosis, accompanied by the release of NETs. NETs enable the entrapment and inactivation of pathogenic microorganisms. Nevertheless, increasing evidence indicates that uncontrolled or excessive production of NETs is associated with the exacerbation of inflammation and the development of autoimmunity [8]. NET is composed of the contents of the cell nucleus, including DNA and granular components such as proteolytic enzymes, including cationic serine proteases: proteinase 3, Cathepsin G (CathG), neutrophilic elastase (NE); myeloperoxidase (MPO); (bactericidal/permeability-increasing protein—BPI); lactoferrins; gelatinases B; cathelicidins (LL-37 or CAP-18; histone proteins (core and linker H1 histones) and tryptases [25]) that are released into the extracellular space [26,27]. Considering the systemic nature of AIH, it seems important to search for some diagnostic indicators for the detection and monitoring of an increased systemic inflammatory response, which is inextricably linked to the course of this disease. The main aim of the study is to assess the diagnostic usefulness of the LDG percentage, including the fraction showing MPO expression, as markers of systemic inflammation in AIH.

## 2. Materials and Methods

### 2.1. Characteristics of Patients

The study included 25 patients treated for AIH in the Department and Clinic of Gastroenterology of the Independent Public Clinical Hospital No. 4 of the Medical University of Lublin. The control group consisted of healthy people, represented by 20 volunteers over 18 years of age (Table 1). The inclusion criteria in this group include the absence of AIH and/or inflammation, based on the correct results of routine laboratory tests for C-reactive protein (CRP) and white blood cells (WBC) (Appendix A). The criterion for inclusion into the study group was the presence of AIH. Both women and men aged 18 and over were included. The exclusion criteria in the study group include chronic viral hepatitis, primary biliary cholangitis (PBC), primary sclerosing cholangitis (PSC), alcoholic or non-alcoholic fatty liver disease, drug-induced liver disease, hepatobiliary infection, hereditary metabolic disease of the liver, and coexistence of any other autoimmune disease. The study design was approved by the Bioethics Committee at the Medical University of Lublin (KE-0254/21/2016). All potential study participants were informed about its course and purposefulness. Written consent was obtained from all participants.

The study group consisted of 22 women and 3 men who met the inclusion criteria for the study. The diagnosis of AIH was based on commonly used guidelines (presence of typical antibodies, elevated serum concentration of immunoglobulin G and characteristic histologic demonstration in liver histology). Any cases of acute or CLD except AIH, or its overlap with PSC and PBC, were perceived as exclusion criteria. Viral and cholestatic liver disorders, together with the presence of clinically significant inflammatory process, were excluded in all participants. 8 patients with AIH were already diagnosed with LC. The diagnosis of LC was based on common criteria [28]. The presence of portal hypertension was proven in the Doppler mode abdominal ultrasound examination (diameter of portal vein ≥13 mm), while other potential reasons for existing portal hypertension were excluded [28]. All of the patients included in the survey achieved clinical and biochemical remission after 4 months from the introduction of treatment. An amount of 10 mL of venous blood was collected from all subjects with disposable equipment. Then, the percentages of LDG, including the fraction showing MPO expression, were determined using flow cytometry. In addition, all patients underwent haematological tests, including assessment of the WBC count, Platelet (PLT) count and biochemical tests, including CRP, Gamma Glutamyl Transpeptidase (GGTP), Alanine Aminotransferase (ALT) and Aspartate Aminotransferase (AST). Based on these parameters, the following indicators were calculated: AAR (AST/ALT ratio), APRI (AST/PLT-ratio index), FIB-4 (fibrosis-4)—(age * AST/PLT * ALT ½), GPR (GGTP/PLT ratio).

The clinical and demographic characteristics of the study participants are presented in Table 1.

### 2.2. Apparatus and Methodology

#### 2.2.1. Test Material

The research material included blood obtained from a vein in the arm. Blood sampling was performed in the morning from fasting patients. Blood was first collected in vacuum clot tubes, where biochemical tests were performed, and then in vacuum tubes containing the anticoagulant K3EDTA (Sarstedt) to perform cytometric determinations of LDG and LDG expressing MPO. The tubes with the material collected “on the clot” were allowed to clot for about 20–30 min, and then centrifuged at the speed of 1000× *g* for 10 min. Whole blood for cytometric tests was analyzed up to 2 h after receiving the material.

#### 2.2.2. Methods

##### Biochemical Determinations

Determinations of biochemical parameters were performed using the COBAS 6000 apparatus, Roche Diagnostics Poland (Warszawa, Poland).

##### Haematological Determinations

Haematological determinations were performed using a Sysmex XN 1500 (Sysmex Group, Warszawa, Poland) apparatus.

##### Isolation of Low-Density Granulocytes

The whole peripheral blood was transferred to a 15 mL Falcon tube and diluted 1:1 with buffered saline without calcium and magnesium ions (PBS, Biomed, Lublin, Poland). The diluted blood was then slowly layered on Histopaque-1077 (Sigma-Aldrich, St. Louis, MO, USA) in a 2:1 ratio (i.e., twice the volume of blood). The reagent used has a density (d = 1.077 g/mL) that allows the separation of the PBMC fraction containing the LDG population from erythrocytes, mature granulocytes and plasma. The samples were centrifuged after layering for 20 min at 1360× *g* at room temperature with the option of slow spinning up and slowing down the rotor in the centrifuge. Then, after centrifugation in the density gradient, the PBMC fraction was obtained, which was in the interface between the plasma layer and the Histopaque reagent used, and, at the very bottom of the test tube, the erythrocyte fraction with granulocytes. At the very beginning, the PBMC fraction, where LDG deposited, was collected. In the next step, these cells were purified by adding 3 mL of a buffered saline solution without calcium and magnesium ions (PBS, Biomed) and centrifuged at 500× *g* for 5 min. The supernatant was removed and the obtained isolate was designated for a cytometric analysis.

##### Viability Analysis of the LDG Population by Flow Cytometry

The assessment of the viability of LDG was performed in a cell isolate that was obtained from the PBMC fraction by centrifugation in a density gradient (Histopaque reagent, Thermo Fisher Scientific, Waltham, MA, USA) prior to cytometric analysis. Propidium iodide (IP) (eBioscience Propidium Iodide, Thermo Fischer Scientific) was used to stain apoptotic cells (PI−). 10 µL of PI was added to 100 µL of LDG isolate (1 × 10^6^ cells), after which the cells were incubated for 5 min. Then 2 mL of buffered saline without calcium and magnesium ions (PBS, Biomed) was added and centrifuged 5 min at 300× *g* in centrifuge (Centrifuge 5810R, Eppendorf, Hamburg, Germany). The supernatant was discarded and the labelled LDG were suspended in 400 µL of a buffered saline solution without calcium and magnesium ions. Shortly thereafter, the samples were cytometrically analyzed. It was shown that the LDG (PI−) population was characterized by similar high viability in the control and study groups (>98%), the minimum viability in both groups was >75%, and the maximum reached 99% (Figure 1).

##### Immunophenotyping

First, the Navios flow cytometer (Beckman Coulter, Brea, CA, USA) was calibrated using three sizes of fluorescent beads (Flow Check Pro Fluorospheres). For the calibration of the blue laser (488 nm) 10 μm spheres were used, for the red laser (635 nm) 6 μm spheres were used and for the violet laser (405 nm) 3 μm spheres were used (the laser signal calibration is presented in Appendix A).

In order to determine the LDG immunophenotype, the obtained cell isolate was divided into two parts. One part of the PBMC fraction was incubated with monoclonal antibodies CD10-PE-Cy5, CD14-PE and CD15-FITC (Becton Dickinson, Franklin Lake, NJ, USA) in an amount of 10 µL for 20 min in the dark at room temperature. The samples were then centrifuged at 300× *g* for 5 min. The supernatant was discarded and the stained LDG subjected to cytometric analysis, the results of which are shown in Figure 2.

The other part of the PBMC fraction was permeabilized with the cell membrane to stain intracellular MPO. A Cell Fixation and Permeabilization Kit (Abcam, Cambridge, UK) was used for permeabilization. For this purpose, the cells were first incubated with monoclonal antibodies staining extracellular antigens. 10 µL of CD10-PE-Cy5 and CD15-FITC monoclonal antibodies were added and incubated for 20 min in the dark at room temperature. Then 3 mL of a buffered saline solution without calcium and magnesium ions was added and centrifuged at 300× *g* for 5 min. The supernatant was discarded and 100 µL of Reagent A, consisting of formaldehyde which fixes the cells, was added to the cell pellet. The cell pellet was then incubated for 15 min at room temperature. In the next step, 3 mL of a buffered saline solution without calcium and magnesium ions was added and centrifuged at 300× *g* for 5 min. The supernatant was discarded and 100 µL of Reagent B was given to the cell pellet, which permeabilizes the cell membrane causing its partial disintegration by creating slits in the cell membrane, which facilitates the penetration of antibodies against intracellular antigens. Next, 10 µL of MPO-PE monoclonal antibody (Becton Dickinson) was added and incubated for 15 min in the dark at room temperature. After this time, 3 mL of a buffered saline solution without calcium and magnesium ions was added and centrifuged at 300× *g* for 5 min. The supernatant was discarded, and 500 µL of buffered saline without calcium and magnesium ions was added to the stained LDG and analyzed on a flow cytometer (Figure 3). Minimum cell flow was used for each analysis and 100,000 cells were counted. Navios Software v. 1.0 and Kaluza Analysis software v. 2.1 (Beckman Coulter) were used to analyze the obtained results.

### 2.3. Statistical Methods

The collected data was analyzed using the Statistica v. 13 PL and MedCalc v 15.8 PL software. The distribution of the categorized data was presented as percentages. The normality of the distribution of continuous data was assessed using the D’Agostino Pearson test. For non-normal distributions, non-parametric tests and medians and interquartile ranges were used as measures of data clustering and dispersion, respectively. The Mann-Whitney *U* test was used to evaluate the differences between continuous variables. The correlation between the variables was assessed using the Spearman’s rank correlation test. In the assessment of the diagnostic usefulness of selected variables (for which statistically significant results were obtained in the Mann-Whitney *U* test or provided these were the parameters not assessed routinely), ROC curve analysis was used to differentiate different clinical conditions. In all the analyses, the results of *p* < 0.05 were considered statistically significant.

## 3. Results

### 3.1. Assessment of the Diagnostic Usefulness of Selected Morphological and Inflammatory Markers, Including the Percentage of LDG and Its Fraction Showing MPO Expression in Detecting AIH

The median CRP was significantly higher in the study group as compared to the control group (1.5 vs. 3 mg/dL; *p* = 0.0157). In patients with AIH, a significantly higher median percentage was observed in the case of: LDG (1.2 vs. 0.1; *p* = 0.0001) and LDG fractions showing MPO expression (0.8 vs. 0.3; *p* = 0.0017), compared to the control group. (Appendix A, Figure 4A,B).

The percentage of LDG was characterised by 100% of sensitivity and 55% of specificity (AUC = 0.84; *p* < 0.0001), while the percentage of MPO-expressing LDG was characterised by 92% of sensitivity and 55% of specificity (AUC = 0.78; *p* = 0.0001) in detecting AIH (Table 2, Figure 5A,B).

### 3.2. Assessment of the Diagnostic Usefulness of Selected Morphological and Inflammatory Markers Including the Percentage of LDG and Its Fraction Showing MPO Expression in the Detection of LC in the Course of AIH

There was an insignificantly higher median percentage of LDG in non-cirrhotic (non-LC) patients as compared to the patients who have had cirrhosis of the liver (LC) (*n* this case trend into significance was noted: 2.1 vs. 0.4 [%]; *p* = 0.0577). Similarly, an insignificantly higher median percentage of LDG fraction expressing MPO (1.4 vs. 0.5 [%]; *p* = 0.3425) in non-cirrhotic patients, as compared to the patients who have had cirrhosis of the liver was noted. There were no statistically significant differences in LDG and LDG fraction expressing MPO in the patients without cirrhosis, as compared to the patients with cirrhosis in the course of AIH. The median values of haematological indices were significantly higher: WBC in the patients without cirrhosis as compared to the group of patients with cirrhosis in AIH (7.2 vs. 4.6 [10^3^/µL]; *p* = 0.0013), PLT (246.0 vs. 85.0 [10^3^/µL]; *p* = 0.0402). Among the biochemical markers, the following indicators were characterized by high median values in the patients with cirrhosis, in comparison with the patients without cirrhosis: GPR (1.9 vs. 1.2; *p* = 0.0085), AAR (1.2 vs. 0.8; *p* = 0.0708), APRI (2.0 vs. 0.6; *p* = 0.0346) and FIB-4 (3.6 vs. 1.3; *p* = 0.0044), (Appendix A).

The values of AUC with the designated cut-off points, as well as the sensitivity and specificity analysis of the percentage of LDG and the LDG fraction showing MPO expression in the detection of LC in the course of AIH, are presented in Table 3. The assessment of the percentage of LDG was characterized by 50% of sensitivity and almost 94% of specificity in the detection of LC in the course of AIH (AUC = 0.75, *p* = 0.0391). On the other hand, the evaluation of the LDG fraction showing MPO expression was characterized by nearly 63% of sensitivity and more than 62% of specificity in the detection of LC in the course of AIH. However, the result was not statistically significant (AUC = 0.62, *p* = 0.3101). WBC was characterized by 100% of sensitivity and 75% of specificity (AUC = 0.91; *p* < 0.0001), while the PLT was characterized by 100% of sensitivity and 81.25% of specificity in the detection of LC in the course of AIH (AUC = 0.95; *p* < 0.0001). The following parameters were characterized by the highest diagnostic value among biochemical indirect markers of liver fibrosis in the detection of LC in the course of AIH: FIB-4 (AUC = 0.87; *p* < 0.0001), APRI (AUC = 0.78; *p* = 0.0054) and AAR (AUC = 0.73; *p* = 0.0349). Of all the assessed biochemical indicators, the FIB-4 indicator was characterized by 100% of sensitivity and 62.5% of specificity in the detection of LC in the course of AIH. The detailed data are included in Table 3 and Figure 5C,D.

### 3.3. Correlation between LDG and Selected Demographic-Clinical and Inflammatory Parameters in the Study Group (AIH)

A statistically significant, moderate, positive correlation was observed between the percentage of LDG and CRP concentration (rho = 0.418; *p* = 0.0420). A significant moderate positive correlation was also observed between the LDG fraction showing MPO expression and the CRP concentration (rho = 0.475; *p* = 0.0189). The detailed data on the correlation between the percentages of LDG in the fraction showing MPO expression and selected demographic, clinical and inflammatory parameters in the study group (AIH) are presented in Table 4.

### 3.4. Correlation between the Percentage of LDG, Including the Fraction Showing MPO Expression with Selected Laboratory Indices Reflecting the Degree of Liver Fibrosis in the Group of Patients with LC

A statistically significant, strong, positive correlation was found between the percentage of the LGD fraction showing MPO expression and APRI. The detailed data on the correlation between the percentage of LDG, including the fraction showing MPO expression, with selected laboratory indicators reflecting the degree of liver fibrosis in the group of patients with LC are presented in Table 5.

## 4. Discussion

In the conducted study, we showed significantly higher percentages of LDG and LDG fraction expressing MPO, the molecular exponents of NETosis in AIH. Our results indicate the role of LDG, including the MPO-expressing fraction, as a potential biomarker in this disease. MPO is a lysosomal enzyme present in large amounts in the azurophilic granules of neutrophils and at low levels in monocyte granules. According to some authors, increased levels of MPO in the bloodstream are associated with inflammation and with increased states of oxidative stress [27]. Increased activation of immune system cells in the course of AIH, found in our study in the form of a higher median LDG percentage, and the LDG fraction showing MPO expression in people with AIH, as compared to the control group, seems to be a consequence of an inflammatory process in the liver. This process is characterized by infiltration of inflammatory cells within hepatocytes [29]. Higher LDG values, including those expressing MPO, indicate ongoing inflammation and possibly ongoing NETosis. The imbalance between the formation and degradation of NETs in the process of NETosis may be related to the pathogenesis of autoimmune diseases. A long-term exposure to NET components is associated with autoimmunity and increases the risk of systemic organ damage [8]. Therefore, the authors of this study decided to check the diagnostic usefulness of selected inflammatory indicators in the development and course of AIH. It turns out that these parameters can become objective indicators for assessing the inflammatory response of the body, the severity of the inflammation developing in the course of AIH and the risk of LC associated with a prolonged activation of the immune system cells. The results indicate that the percentage of LDG and LDG expressing MPO were more sensitive in detecting AIH, in contrast to the routinely used markers of inflammation, such as WBC and CRP.

Following the conclusions described, it was decided to evaluate the percentage of LDG and the LDG fraction showing MPO expression in patients with AIH, as compared to the control group. To the best of our knowledge, this is the first study regarding this issue in AIH. The results of our study showed statistically significantly higher percentages of LDG and LDG fraction showing the expression of MPO in patients with AIH, as compared to the control group. The obtained results correspond perfectly with literature reports on other autoimmune diseases [11,17,19,30]. Our study explored LDG as an indicator of the inflammatory background of AIH. The secondary aim of the study was to evaluate LDG as an indicator of LC, thus we decided to assess routinely used serological and calculated markers of liver fibrosis (AAR, APRI, FIB-4 and GPR) and to correlate them with LDG. To the best of our knowledge, similar investigations have not been performed, so far. Interestingly, a positive correlation between LDG MPO+ and APRI was noted. APRI constitutes a well-known indirect parameter of liver fibrosis, commonly used in the non-invasive assessment of patients with chronic liver disorders. Its diagnostic accuracy and utility differ according to certain liver pathology; APRI is often used in patients with HBV- and HCV-related chronic liver disease. In the course of AIH, APRI seems to be perceived as a marker with average diagnostic accuracy, potentially helpful in the exclusion of liver fibrosis. The observed relationship between LDG MPO+ and APRI suggests that assessment of the diagnostic role of LDG in the process of liver fibrosis could be promising, and further research on this issue is warranted. Further studies should concern histopathological evaluation of liver fibrosis in AIH patients within the context of measurement of LDG level [31,32,33].

Zhang et al. suggested a significant relationship between the increased level of LDG and inflammation. NET-forming LDGs can lower the activation threshold of T lymphocytes present in the PBMC fraction in patients with the above-mentioned diseases and enhance their immune response. An increased percentage of LDG can spontaneously form NETs, which may affect T-cell cluster formation, an increased expression of CD25 and CD69 lymphocyte activation markers, and phosphorylation of ZAP70 signalling kinase associated with the T cell receptor (TCR) in CD4+ T cells [22].

Lin et al. observed significantly higher percentages of LDG in the blood of patients with psoriasis (*n* = 15), as compared to the control group (*n* = 6), (34.6 ± 5.1% vs. 11.0 ± 2.0%, *p* < 0.01). These studies suggest that neutrophils and LDG may play a significant role in the course of psoriasis by releasing NET and its components, including IL-17, MPO, EN. Moreover, they suggest that circulating neutrophil populations, including LDG, are more likely to undergo NETosis [19].

In their research Cloke et al. also isolated the LDG fraction in 21 HIV patients. Their results indicate that the prevalence of LDG is significantly higher in HIV seropositive patients with low levels of CD4+ T cells (*n* = 10), as compared to patients with high numbers of CD4+ T cells (*n* = 11). Additionally, the studies have shown some differences in the NDG and LDG phenotypes. LDG is characterized by an increased expression of the markers: CD11b, CD15, CD33, CD66b, and CD63, and a decreased CD16 and arginase 1 [17]. It should be noted that both the available literature and the results of our study clearly indicate increased levels of LDG and LDG MPO+ fraction in the PBMC as potential markers among patients with autoimmune disorders [34]. AIH was poorly explored in this field and our study fills this gap. However, further investigations should evaluate the concentration of LDGs in both serum and liver biopsy specimens of AIH patients, with a comparison of achieved results. Furthermore, the concentration of LDGs should be evaluated at the diagnosis of AIH and after the introduction of pharmacological treatment in patients with a response to the therapy. Our study constitutes a pilot one performed on a relatively small number of patients. Nevertheless, the first data are quite promising. The examined granulocyte subpopulations are likely to play an active role in the development and maintenance of inflammation in AIH. Cytometric evaluation of the LDG percentage, including the LDG MPO+ fraction in patients with AIH, may contribute to the improvement of the diagnostic interpretation of the patient’s test results. They can be potentially useful markers in more precise diagnosis of AIH, as well as in determining its severity. The obtained data will certainly allow for the diagnosis of inflammation in the course of AIH. However, they are only auxiliary markers of the inflammatory reaction. Additionally, the obtained results allow us to conclude that there is a certain potential role of the above markers in the non-invasive assessment of the degree of LC.

## 5. Conclusions

Our study indicates that the assessment of the percentage of LDG, including the fraction showing MPO+ expression, is characterized by very high sensitivity (100% and 92%, respectively) and moderate specificity (55% in both cases) in detecting AIH. The usefulness of the analyzed markers in the diagnosis of LC in the course of AIH is also noteworthy. We showed that the evaluation of the percentage of LDG and the LDG fraction showing MPO expression is characterized by moderate sensitivity (50% and 62.75%, respectively) and high specificity (93.75% and 62.50%) in the detection of LC in the course of AIH. Therefore, the assessment of the LDG percentage and the LDG MPO+ fraction after an appropriate validation may become useful markers in the diagnosis of AIH. In the future, the use of non-invasive laboratory LC diagnostics may reduce the number of performed liver biopsies, and, thus, reduce the risk of complications associated with this procedure.

## Figures and Tables

**Figure 1 jcm-11-02174-f001:**
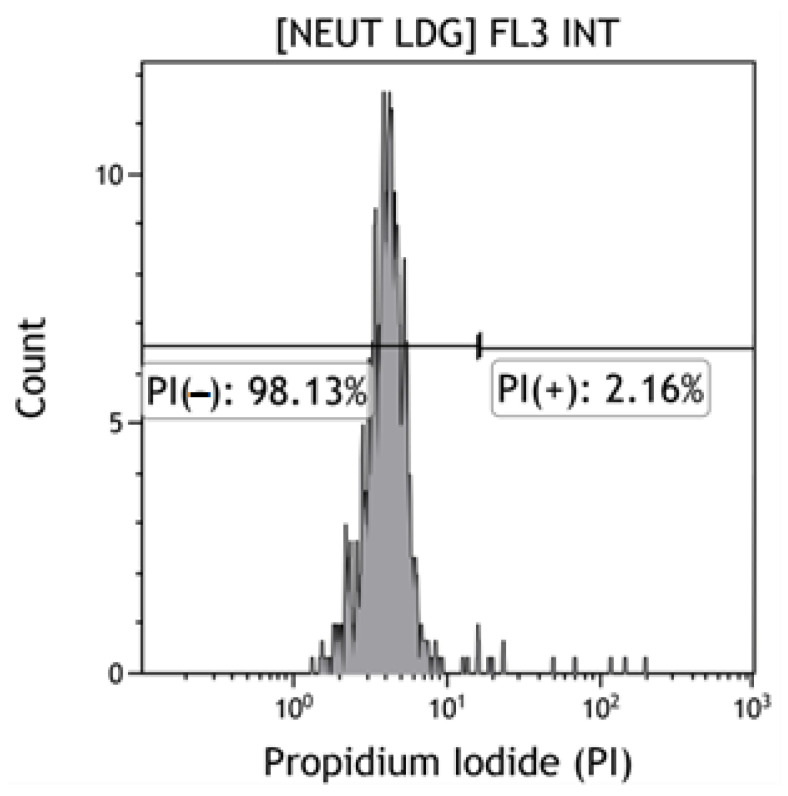
Viability of the LDG population. **A-C** Cytograms. The “PI−” gate is the IP negative LDG population—the percentage of viable cells. The “PI+” gate is the population of dead cells (my own source). [Neut LDG] FL3 INT—[Low density granulocytes], a fluorescence channel on which the fluorescence of propidium iodide is induced.

**Figure 2 jcm-11-02174-f002:**
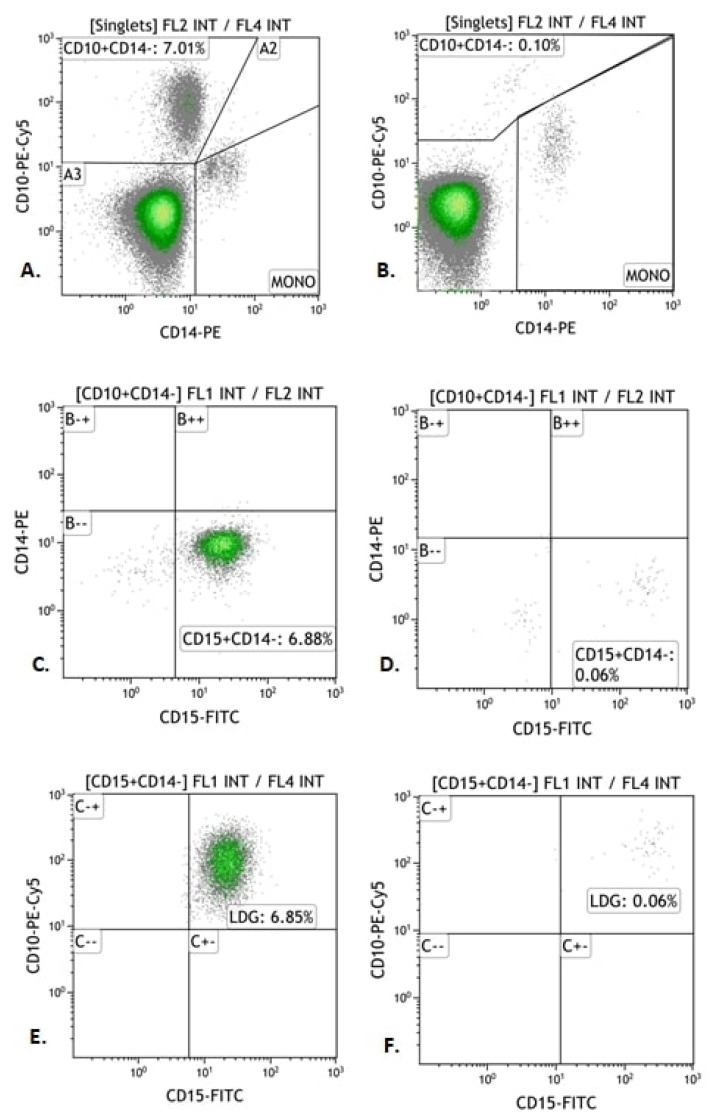
Exemplary Immunophenotyping of LDG (CD10+ CD15+ CD14−) in patients with autoimmune hepatitis (AIH) and in the control group. (**A**,**B**) Cytograms. Percentage of cells positive for CD10 antigen and negative for CD14 (“CD10 Gate + CD14−”). (**C**,**D**) Cytograms. Percentage of LDG positive for CD15 antigen and negative for CD14 antigen (“CD15+ CD14− Gate”). (**E**,**F**) Cytograms. Percentage of LDG positive for CD15 and CD10 antigens and negative for CD14 (“LDG Gate”). (**A**,**C**,**E**) Cytograms—study group; (**B**,**D**,**F**) Cytograms—control group.

**Figure 3 jcm-11-02174-f003:**
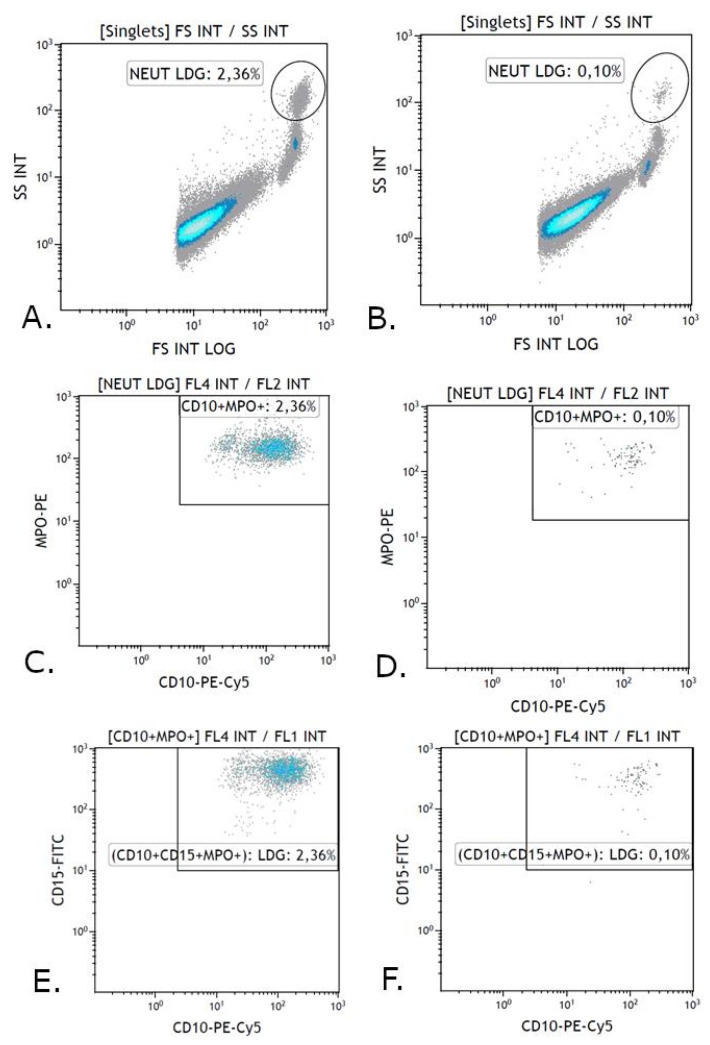
Exemplary analysis of the immunophenotype of LDG (CD10+ CD15+) with the expression of MPO+ antigen in patients with autoimmune hepatitis and in the control group. (**A**,**B**) Cytograms. Percentage of single cells relative to FSC detector and SSC (“NEUT LDG Gate”). (**C**,**D**) Cytograms. Percentage of cells positive for CD10 antigen and MPO (“CD10+ MPO+ Gate”). (**E**,**F**) Cytograms. Percentage of LDG positive for antigens: CD10, CD15, MPO+ (“CD10+ CD15+ MPO+ Gate”). (**A**,**C**,**E**) Cytograms—study group. (**B**,**D**,**F**) Cytograms—control group.

**Figure 4 jcm-11-02174-f004:**
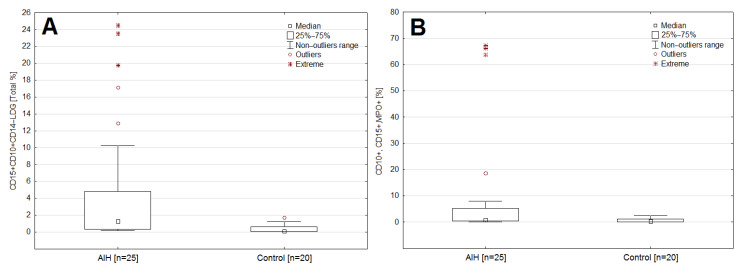
Box-whisker plot comparing the percentage of LDG (**A**) and the LDG fraction (**B**) showing MPO expression in the AIH group and the control group.

**Figure 5 jcm-11-02174-f005:**
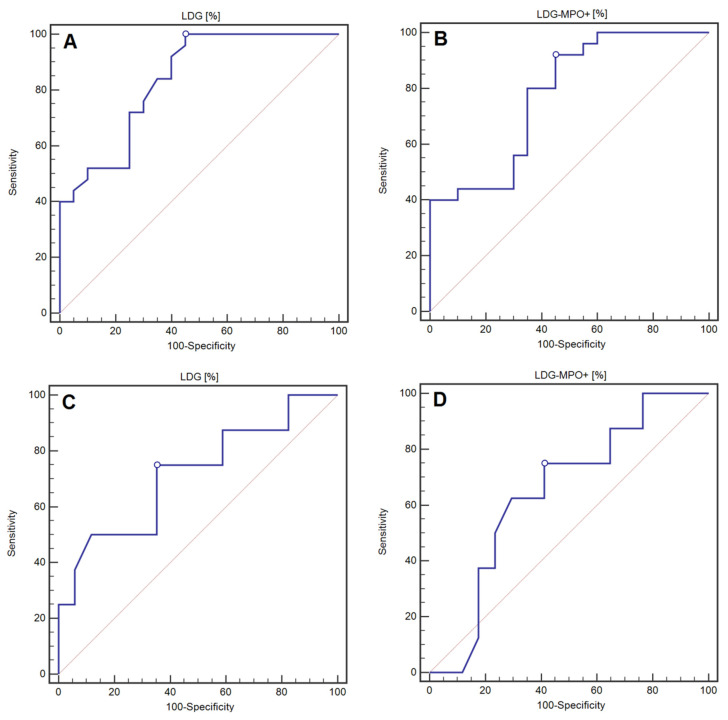
ROC curves representing the assessment of the diagnostic usefulness of the percentage of LDG (**A**) and the LDG fraction with the expression of MPO (**B**) in the detection of AIH. ROC curves representing the assessment of the diagnostic usefulness of the percentage of LDG (**C**) and the LDG fraction with the expression of MPO (**D**) in the detection of LC in the course of AIH.

**Table 1 jcm-11-02174-t001:** Characteristics of the study group and control group.

Demographic Data			
Variable		Study Group (AIH) *n* = 25 (%)	Control Group *n* = 20 (%)
1.	Gender		
	Women	22 (88%)	18 (90%)
	Men	3 (12%)	2 (10%)
2.	Age [years]		
	Median (range)	56 (23–80)	43.5 (21–69)
3.	BMI [kg/m^2^]		
	Median (range)	25.5 (18.7–37.1)	21.0 (17–29)
Clinical data			
4.	Duration of the disease [years]		
	Median (range)	13 (1–25)	-
5.	Treatment		
	Steroids	16 (64%)	-
	Immunosuppressants	1 (4%)	-
	Steroids + Immunosuppressants	8 (32%)	-
6.	Family history towards AIH		
	Negative	19 (76%)	-
	Positive	6 (24%)	-
7.	LC	8(32%)	-
	Non-LC	17 (68%)	-
8.	Comorbidities		
	Yes *	13 (52%)	6 (30%)
	No	12 (48%)	13 (70%)

AIH—autoimmune hepatitis, BMI—body mass index, LC—liver cirrhosis. * including: nephrolithiasis, osteoporosis, bronchial asthma, arterial hypertension, diabetes, glaucoma, chronic heart failure.

**Table 2 jcm-11-02174-t002:** Assessment of the diagnostic usefulness of selected inflammatory markers and the percentage of LDG, including the fraction showing MPO expression in the detection of AIH.

Variable	Sensitivity (%)	Specificity (%)	Cut-Off Point	AUC [95%CI]	*p*
WBC [10^3^/µL]	44	90	>6.94	0.63 [0.47–0.77]	0.1375
CRP [mg/L]	83.3	55	>1.5	0.71 [0.56–0.84]	0.0075 *
LDG [%]	100	55	>0.10	0.84 [0.70–0.93]	<0.0001 *
LDG MPO+ [%]	92	55	>0.31	0.78 [0.63–0.89]	0.0001 *

WBC—White blood cells, CRP—C-reactive Protein, LDG—Low Density Granulocytes, MPO—Myeloperoxidase, *—Statistically significant result.

**Table 3 jcm-11-02174-t003:** Assessment of the diagnostic usefulness of selected biochemical and inflammatory markers, including the percentage of LDG and the LDG fraction showing MPO expression in the detection of LC in the course of AIH.

Variable	Sensitivity (%)	Specificity (%)	Cut-Off Point	AUC [95%CI]	*p*
AST [IU/L]	87.5	47.06	>42	0.57 [0.35–0.76]	0.5834
ALT [IU/L]	100	41.18	≤101	62.0 [0.41–0.81]	0.2825
GGTP [IU/L]	87.5	47.06	≤122	0.63 [0.42–0.81]	0.2767
GPR	87.5	31.25	>0.4	0.58 [0.36–0.77]	0.5590
AAR	87.5	56.25	>0.84	0.73 [0.52–0.89]	0.0349 *
APRI	87.5	68.75	>0.93	0.78 [0.56–0.92]	0.0054 *
FIB-4	100	62.5	>1.47	0.87 [0.67–0.97]	<0.0001 *
WBC [10^3^/µL]	100	75	≤6.2	0.91 [0.73–0.99]	<0.0001 *
PLT [10^3^/µL]	100	81.25	≤201.00	0.95 [0.78–0.99]	<0.0001 *
CRP [mg/mL]	71.43	11.76	>1.3	0.51 [0.30–0.72]	0.9537
LDG [%]	50	93.75	≤0.25	0.75 [0.53–0.90]	0.0391 *
LDG MPO+ [%]	62.75	62.50	≤0.59	0.62 [0.41–0.81]	0.3101

AAR—Aspartate aminotransferase—to-Alanine aminotransferase Ratio, APRI—Aspartate aminotransferase-to-Platelet Ratio Index, ALT—Alanine Aminotransferase, AST—Aspartate Aminotransferase, AUC—Area under curve, CD—Cluster of differentiation, CI—Confidence Interval, CRP—C-Reactive Protein, FIB-4—Fibrosis-4, GPR—Gamma-glutamyl-transpeptidase-to-Platelet Ratio, GGTP—Gamma Glutamyl Transpeptidase, LDG—Low Density Granulocytes, MPO—Myeloperoxidase, PLT—Platelet, WBC—White Blood Cells, *—statistically significant result.

**Table 4 jcm-11-02174-t004:** Correlation between LDG and selected demographic-clinical and inflammatory parameters in the study group (AIH).

Variable	LDG Rho *p*	LDG MPO+ Rho *p*
Age [years]	−0.069 0.7417	0.101 0.6321
Duration of the disease [years]	0.081 0.6987	0.212 0.3090
BMI [kg/m^2^]	0.004 0.9847	−0.053 0.8023
CRP [mg/L]	0.418 0.0420 *	0.475 0.0189 *
WBC [10^3^/µL]	0.375 0.0647	0.017 0.9374

BMI—body mass index, CRP—C-Reactive Protein, LDG—Low Density Granulocytes, MPO—Myeloperoxidase, WBC—White Blood Cells, *—Statistically significant result.

**Table 5 jcm-11-02174-t005:** Correlation between the percentage of LDG, including the fraction showing MPO expression with selected laboratory indices reflecting liver fibrosis in the group of patients with LC.

Variable	LDG Rho *p*	LDG MPO+ Rho *p*
GPR	0.238 0.5702	0.571 0.1390
AAR	0.214 0.6103	−0.0952 0.8225
APRI	0.000 1.000	0.762 0.0280 *
FIB-4	−0.190 0.6514	0.524 0.1827

AAR—Aspartate aminotransferase–to-Alanine Aminotransferase Ratio, APRI—Aspartate Aminotransferase-to-Platelet Ratio Index, FIB-4—Fibrosis-4, GPR—Gamma-glutamyl-transpeptidase-to-Platelet Ratio, LC—Liver cirrhosis, LDG—Low Density Granulocytes, MPO—Myeloperoxidase, *—Statistically significant result.

## Data Availability

The data used in this study are sensitive patient data, partly belonging to Independent Public Clinical Hospital No. 4 in Lublin. For this reason, they can only be shared with the consent of the hospital. In this case, please contact the hospital’s data protection officer or the director of the hospital for medical affairs (ul. Jaczewskiego 8, 20-954 Lublin, szpital@spsk4.lublin.pl).

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
