# Peer review of "Indicator of Inflammation and NETosis—Low-Density Granulocytes as a Biomarker of Autoimmune Hepatitis"

_jcm, 2022, doi:10.3390/jcm11082174_

Round 1

Reviewer 1 Report

Domerecka et al set out an experiment to assess potential of the presence of CD10+/CD14-/CD15+ low-density granulocyte (LDGs) in PBMC for AIH diagnosis, with plausible mechanistical background on which systemic innate inflammation may be coupled with liver pathology of AIH.

Their hypothesis is worthwhile to be evaluated if the presence of LDGs is pathogenically specific in AIH comparing to disease-controls, as a complementary measure in the diagnosis of AIH. On the contrary, they only focused to compare diagnostic usefulness of LDGs/LDGs-MPO+ with those of many unspecific inflammatory blood markers. 

Concerns

  1. Though LDGs and NETosis might be involved in the pathogenesis of AIH, they are not established requirement of diagnostic AIH pathology in the liver.
  2. Introduction and Discussion are very redundant, not appropriately focusing to AIH.
  3. In Results: Are WBC and CRP worthwhile to be evaluated as diagnostic measures for AIH?
  4. No clinical information was available with regard to the treatment response by steroids/immunosuppressants, that is biochemical remission.
  5. Was the percentage of LDGs in PBMC significantly associated with biochemical remission?
  6. The percentage of LDGs in PBMC of AIH at initial diagnosis should be evaluated as well.
  7. What would be the mechanism for the decrease in LDGs in PMMC of AIH-LC?
  8. In Discussion, Page 16, lane 506: The available literature documenting for the increased levels of LDGs and LDGs-MPO+ in AIH should be referred appropriately.

Author Response

Domerecka et al set out an experiment to assess potential of the presence of CD10+/CD14-/CD15+ low-density granulocyte (LDGs) in PBMC for AIH diagnosis, with plausible mechanistical background on which systemic innate inflammation may be coupled with liver pathology of AIH.

Their hypothesis is worthwhile to be evaluated if the presence of LDGs is pathogenically specific in AIH comparing to disease-controls, as a complementary measure in the diagnosis of AIH. On the contrary, they only focused to compare diagnostic usefulness of LDGs/LDGs-MPO+ with those of many unspecific inflammatory blood markers.

Concerns

  1. Though LDGs and NETosis might be involved in the pathogenesis of AIH, they are not established requirement of diagnostic AIH pathology in the liver.

It was the first study on LDGs in the context of AIH. We did not aim to present them as evident markers of the disease. According to the inflammatory background of AIH and available results concerning LDGs in other autoimmune pathologies, we decided to investigate their potential role in AIH patients. For now we need to get more detailed data on this topic, in order to predict the potential role of LDGs in the diagnosis of AIH.

  1. Introduction and Discussion are very redundant, not appropriately focusing to AIH.

The sections were modified and all the redundant information has been removed, while a special attention has been given to the background of AIH, according to the suggestion of the Reviewer.

  1. In Results: Are WBC and CRP worthwhile to be evaluated as diagnostic measures for AIH?

They seem to be valuable additional parameters of AIH in the scope of coexisting inflammation. It was the main reason for which we decided to include them to the tested parameters. In general, they are not specific for AIH, but we used them because of LDGs, trying to broaden the context of inflammation.

  1. No clinical information was available with regard to the treatment response by steroids/immunosuppressants, that is biochemical remission.

All of the patients included in the survey achieved remission after the introduction of pharmacological treatment. The information about the positive response on the treatment was included in the manuscript. Our aim was not to evaluate the examined hematological markers with regard to the severity of AIH or the outcome of its treatment - it could be the topic of another research paper and even it should become one for the scientific purpose. Probably, in the future, we will evaluate the concentration of LDGs in both: serum and liver biopsy specimens and compare the results. However, in this manuscript we performed just a single assessment of LDGs concentration in the serum of AIH patients.

  1. Was the percentage of LDGs in PBMC significantly associated with biochemical remission?

The aim of our study was not to associate a certain percentage of evaluated cell subsets with a particular stage of AIH or the response to the pharmacological treatment. It was the first investigation in our unit concerning the assessment of LDGs in the course of AIH. So the group of patients included in the study underwent just a single assessment of LDGs in the serum.                                                                                         

  1. The percentage of LDGs in PBMC of AIH at initial diagnosis should be evaluated as well.

This suggestion is very appropriate and worth implementing in the future studies. AIH patients enrolled in our study were not homogenous according to the duration of AIH; they were enrolled in the study in the different stages of AIH. We did not aim to evaluate the baseline concentration of LDGs at diagnosis of AIH - it is a valuable idea for the next article.

  1. What would be the mechanism for the decrease in LDGs in PMMC of AIH-LC?

It seems to be related to the decreased severity of inflammation in case of more advanced liver fibrosis. The first stages of this profibrotic cascade are more likely to be inseparably connected with inflammation and the activation of cytokines together with growth factors.

  1. In Discussion, Page 16, lane 506: The available literature documenting for the increased levels of LDGs and LDGs-MPO+ in AIH should be referred appropriately.

A new reference, referring to this paragraph, was included.

  1. Moderate English changes required

Thank you for your remark. The entire text of the manuscript was checked for English errors. All errors have been corrected.

Reviewer 2 Report

LDG and LDG MPO+ seemed to be not as good as WBC and PLT for detecting LC in AIH because of moderate sensitivity. Can LDG/LDG MPO+ combining with WBC or PLT have a better diagnostic sensitivity than WBC or PLT alone for detecting LC in AIH?

Author Response

LDG and LDG MPO+ seemed to be not as good as WBC and PLT for detecting LC in AIH because of moderate sensitivity. Can LDG/LDG MPO+ combining with WBC or PLT have a better diagnostic sensitivity than WBC or PLT alone for detecting LC in AIH?

We did not find this dependency in our results. Nevertheless, combining various inflammatory markers in a common diagnostic panel and its application in a larger group of AIH patients could potentially increase their individual diagnostic accuracy. Probably, it will be the next step in our future investigations.

Round 2

Reviewer 1 Report

Domerecka et al revised the manuscript according to the Reviewer’s comments. Nevertheless, their hypothesis should be again evaluated whether the presence of LDGs in PBMC is pathogenically specific in AIH comparing to disease-controls, eg., NASH, as a complementary measure in the diagnosis of AIH. Comparison between AIH in remission to healthy volunteer is necessary and interesting to have implication about unresolved stigmata of systemic inflammation in AIH, but not appropriate to access clinical relevance of LDGs for AIH.

Concerns:

1. Discussion is still redundantly described, with plenty of liver-unrelated literatures about DM, PM, and SLE.

Author Response

Concerns

  1. Discussion is still redundantly described, with plenty of liver-unrelated literatures about DM, PM, and SLE.

Thank you for your remark. The sections were modified and all the redundant information has been removed, according to the suggestion of the Reviewer.

This manuscript is a resubmission of an earlier submission. The following is a list of the peer review reports and author responses from that submission.